# Molecular Epidemiology of Group B Streptococcus Colonization in Egyptian Women

**DOI:** 10.3390/microorganisms11010038

**Published:** 2022-12-22

**Authors:** Sarah Shabayek, Verena Vogel, Dorota Jamrozy, Stephen D. Bentley, Barbara Spellerberg

**Affiliations:** 1Department of Microbiology and Immunology, Faculty of Pharmacy, Suez Canal University, Ismailia 41522, Egypt; 2Institute of Medical Microbiology and Hygiene, University Hospital Ulm, 89081 Ulm, Germany; 3Parasites and Microbes, Wellcome Sanger Institute, Hinxton, Cambridge CB10 1SA, UK

**Keywords:** *Streptococcus agalactiae*, Group B Streptococcus, GBS, Egypt, JUNO, molecular epidemiology, MLST, colonization, woman

## Abstract

(1) Background: *Streptococcus agalactiae* or Group B Streptococcus (GBS) causes severe neonatal infections with a high burden of disease, especially in Africa. Maternal vaginal colonization and perinatal transmissions represent the common mode of acquiring the infection. Development of an effective maternal vaccine against GBS relies on molecular surveillance of the maternal GBS population to better understand the global distribution of GBS clones and serotypes. (2) Methods: Here, we present genomic data from a collection of colonizing GBS strains from Ismailia, Egypt that were sequenced and characterized within the global JUNO project. (3) Results: A large proportion of serotype VI, ST14 strains was discovered, a serotype which is rarely found in strain collections from the US and Europe and typically not included in the current vaccine formulations. (4) Conclusions: The molecular epidemiology of these strains clearly points to the African origin with the detection of several sequence types (STs) that have only been observed in Africa. Our data underline the importance of continuous molecular surveillance of the GBS population for future vaccine implementations.

## 1. Introduction

*Streptococcus agalactiae* or Group B Streptococcus (GBS) is a leading cause of neonatal sepsis, pneumonia, and meningitis [1,2]. Recently, the World Health Organization (WHO) [3] indicated a high burden of GBS disease, including stillbirth and preterm birth conditions. The burden of GBS disease is particularly high in Africa, while at the same time, GBS epidemiological data are lacking for many countries of this region [3]. GBS is a pathobiont that colonizes the gastrointestinal and genitourinary tracts of healthy women, and maternal carriage is the major route of GBS transmission to newborn babies [2]. It may also cause serious infections in pregnant and nonpregnant adults [1,2]. As an opportunistic commensal, pathogen GBS reverts from asymptomatic carriage to an invasive state, causing severe infections [1].

Determination of capsular serotype is the classical epidemiological method used for GBS typing [4]. It divides GBS strains into 10 serotypes (Ia, Ib, II–IX) based on the different sialic acid-rich capsular polysaccharides (CPS) [4]. Multi-locus sequence typing (MLST) is also used to classify GBS strains based on allelic variation within seven housekeeping genes [5]. It allows comparisons among different geographic areas [4] and shows a correlation of clonal complexes (CCs) with capsular serotypes [4]. Some CCs are more likely to cause invasive disease, whereas others are mostly associated with asymptomatic carriage. The CC1, CC10/CC12, CC19, and CC23 are well-adapted vaginal commensals with limited invasive potential in neonates [5,6,7,8]. On the contrary, CC17 isolates, in particular ST-17 serotype III, are often associated with neonatal sepsis and meningitis [5,6,9,10]. MLST can reveal geographic restrictions, CC26, for example, is a common lineage of human serotype V isolates in Africa, which has been rarely detected in other regions of the world [6,11,12].

Screening and intrapartum antibiotic prophylaxis (IAP) are recommended effective therapy for pregnant women GBS carriers [2]. However, such preventive strategies are rarely adopted in underdeveloped regions, particularly Africa [13,14]. Vaccines seem to be an alternative and cost-effective option [14]. The population structure of GBS can vary between geographic regions [2], a major aspect to be considered in vaccine development. GBS vaccination among pregnant women has been announced as a priority for the World Health Organization (WHO) in low- and middle-income countries [15]. At present, there is not yet a licensed GBS vaccine available [3,16]. Most ongoing clinical trials employ trivalent and hexavalent serotype-based vaccines, which means that they are dependent on reliable data about the molecular epidemiology of circulating GBS isolates [17,18,19]. Epidemiologic studies characterizing local GBS strains are scarce in Egypt. Molecular epidemiology of GBS strains from vaginal carriage in Ismailia, Egypt, and their possible correlations with capsular serotypes and surface proteins were investigated using whole genome sequencing (WGS) within the JUNO project [20]. Ninety GBS isolates collected from pregnant (n = 56) and nonpregnant (n = 34) women in an earlier investigation were used in the current study [21].

## 2. Materials and Methods

Genomic DNA was extracted from isolates using the GenElute Bacterial Genomic DNA Kit (Sigma-Aldrich, St. Louis, MO, USA) according to the manufacturer’s instructions. Whole-genome sequencing (WGS) was performed at the Wellcome Sanger Institute as part of the global GBS surveillance study JUNO (https://www.gbsgen.net/ (accessed on 17 December 2022)) [20]. Briefly, WGS libraries were prepared using the NEB UltraII library kit on an Agilent Bravo liquid handling platform, and the sequencing was performed on the Illumina HiSeq platform with 150 base-pair read length. Short-read data are available from the European Nucleotide Archive (see Appendix A for sample accession numbers).

### 2.1. MLST Types

MLST of 90 strains were obtained from the genome data. BURST analysis was used to group isolates into the same CC if sharing six out of the seven alleles (single-locus variants) [22]. Each CC was named after its ancestor ST or the predominant ST within this clone [22,23]. 

### 2.2. Pili and GBS Surface Protein Genes

The genome data were used to search for, pilus island (PI) variants, the presence or absence of surface protein genes, including the hypervirulent GBS adhesin *hvgA*/bibA (gbs2018), Alpha-like protein family (*alpha-C*, *rib*, *alp1*, *alp2/3*, *epsilon*), and serine-rich repeat (Srr) protein genes. Identified genes were compared to query sequences, and only results with >95% sequence identity were positive. Sequences for comparison were obtained from McGee et al., 2021 [24]. 

To perform genomic relationship analysis of CC1, the respective genomes were uploaded to REALPHY 1.13 (https://realphy.unibas.ch/realphy/ (accessed on 17 December 2022)) [25]. This online tool constructs a phylogenetic tree by using bowtie2.

## 3. Results

### 3.1. MLST Types

A total of 19 unique MLST STs were identified, of which ST-1 (n = 26) was the most prevalent, accounting for 28.8% of the isolates, followed respectively by ST-14 (n = 10, 11.1%), ST-19 (n = 9, 10%), ST-12 (n = 8, 8.8%), ST-932 (n = 7, 7.7%), ST-569 (n = 6, 6.6%), ST- 23 (n = 5, 5.5%), ST-4 (n = 4, 4.4%), and ST-28 (n = 3, 3.3%). The rest of the STs found (3, 6, 10, 24, 25, 196, 486, 556, 934) were represented by up to two isolates each. For two isolates, no specific ST was found in the public MLST database since these carried a novel allele. Following the report of this allele to the MLST database, the new ST profile 1954 was assigned to both strains. 

The new ST-1954 was found to be a single locus variant of ST-17, as resolved by Burst analysis. Consequently, the corresponding isolates were clustered under CC17. However, ST-17 was not detected in our collection.

The resolved STs were clustered into 6 CCs (CC1, CC4, CC12, CC17, CC19, CC23) and 6 singleton STs (ST-6, ST-24, ST-196, ST-486, ST-569, ST-934). CC1 was the most prevalent (36/90, 40%) due to a high number of isolates representing ST-1 and ST-14. This was followed by CC19 (12/90, 13.3%), CC4 (12/90, 13.3%), CC12 (9/90, 10%), CC23 (7/90, 7.7%), and finally, CC17 (2/90, 2.2%) (Figure 1). Among the singleton strains that could not be assigned to a major CC, ST-569 was the most prevalent singleton (6/90, 6.6%).

### 3.2. Pili and GBS Surface Protein Genes

All GBS genomes were also screened for the pilus island (PI) gene profile, serine-rich repeat Srr protein variant genes, and the *hvgA*/*bibA* adhesion molecule genes (Table 1). Pilli are crucial for GBS colonization, biofilm formation, translocation, and invasion [26]. Three pilus islands have been reported in GBS: PI-1, PI-2a, and PI-2b [26,27,28,29]. All GBS isolates are known to contain a single variant or a combination of two pilus islands [26]. Two-thirds of the isolates we investigated (61/90, 67.7%) contained the PI-1+2a combination, one-third of the isolates (26/90, 28.8%) harbored the PI-1+2b combination, and three isolates contained PI-2a alone. Interestingly, most strains with the PI-I+2b combination belonged to ST-14 (10/10), ST-4 (4/4), ST-932 (6/7), and the two novel ST-1954 isolates. 

Most of our isolates (84/90, 93.3%) harbored the Srr-1 variant gene, while two isolates harbored the Srr-2 variant gene [30,31]. Interestingly, the latter were the two novel ST-1954/CC17 strains. Four isolates did not harbor any Srr protein variants genes, all of which were serotype II. Another common surface protein is BibA, an essential immunogenic adhesin that is highly conserved in GBS [32,33]. The well-characterized HvgA is a BibA variant (gbs2018C), which was defined to be specific to hypervirulent ST-17 isolates [10]. Three isolates in our collection were positive for the HvgA adhesin gene: The two novel ST-1954/CC17 strains and the ST-934 isolate. Hence, as expected, ST-1954/CC17 isolates were typically serotype III, positive for the HvgA and Srr-2 protein genes and contained the PI-1+2b pilus island profile as previously reported for the hypervirulent CC17 [9,10,26,34,35]. Surprisingly, all of the ten ST-14 strains in our collection lacked a functional BibA (gbs2018) adhesin gene due to a premature stop codon. 

We observed an overall clustering of dominant STs into defined serotypes (Table 1, Figure 2). These included serotype V strains with ST-1 (21/28, 75%), serotype VI with ST-14 (9/11, 81.8%), serotype III with ST-19, ST-23, and ST-1954 (11/14, 78.6%), serotype Ib with ST-12 (5/7, 71.4%), serotype Ia with STs 4 and 932 (7/13, 53.8%), and serotype II with STs 28, 569, and 932 (11/16, 68.8%). 

An overall clustering of dominant STs with defined Alpha-like surface protein genes has been observed as well (Table 2, Figure 3). The Alp3 protein gene was exclusively found in ST-1 (22/22, 100%). The Alp2 protein gene was dominant in ST-23 strains (5/10, 50%), the Alpha-C protein gene was found to be associated with ST12 and ST-569 (12/17, 70.6%), Epsilon protein gene with ST-14, ST-19, and ST-4 (18/24, 75%), and finally Rib protein gene with ST932, ST-19, ST-28, and the novel ST1954 (12/15, 80%). The most predominant genetic lineages we could detect were ST-1/V/Alp3/PI-1+2a (20/90, 22.2%) and ST-14/VI/Epsilon/PI-1+2b (9/90, 10%). 

The genotypic profiles indicated a clonal distribution of isolates within our collection. As such, STs were most often associated with a single serotype and a single surface protein gene (Figure 2 and Figure 3). Among the ST-1 and ST14 strains, ST-1/V/Alp3 (20/26) and ST-14/VI/Epsilon (9/10) were, respectively, the predominant lineages. Further genomic analysis showed distinct close clustering of ST-14 strains, which set them apart from ST-1 strains (Figure 4). Homogeneity was also observed for ST-569 isolates, which were all ST-569/II/Alpha-C (5/6), ST-12, which was predominately found to be ST-12/Ib/Alpha-C (5/8), and ST-19 strains, which were predominately expressed either as ST-19/V/Epsilon (5/9) or ST-19/III/Rib (3/9). In addition, all the ST-4 isolates were ST-4/Ia/Epsilon (4/4), ST-23 isolates were exclusively found to be ST-23/III/Alp2 (5/5), and ST-28 isolates exclusively appeared as ST-28/II/Rib (3/3). 

## 4. Discussion

Maternal colonization is a prerequisite for GBS transmission to the newborn infant. Vaginal carriage may be considered a reservoir of strains with diverse serotypes, virulence factors, and antibiotic resistance genes [1]. The overall population structure is similar to that of the United States and Europe [36]. However, some specific local features have been identified. 

In our study, ST-1, ST-19, and ST-23 were the most predominant clones in their corresponding CCs. CC1 and CC19 were the dominant clonal complexes in our collection, representing more than 50% of the analyzed isolates (48/90). With respect to pregnancy, CC1 represented 42% (24/56) of pregnant women isolates versus 35% (12/ 34) among nonpregnant. However, CC19 represented 10.7% (6/56) of pregnant women isolates compared to 17.6% (6/34) among nonpregnant. Consistent reports pointed out the domination of CC1, CC19, and CC23 among asymptomatic pregnant women [5,6,7,8]. The ST-1, ST-19, and ST-23, subtypes of CC1, CC19, and CC23, have been previously reported as well-adapted vaginal commensals displaying a poor invasion ability [5,6,7,8]. 

The Srr proteins are surface-associated structures that bind to fibrinogen and are important in promoting GBS attachment and colonization of the vaginal tract [30,31]. Two variants of the Srr proteins have been described in GBS [30,31]. Two of our serotype III isolates were identified as a novel sequence type, ST-1954, a subtype of CC-17 harboring the hypervirulent adhesin HvgA gene, the serine-rich repeat protein Srr2 variant gene and PI-1+2b pilus subunits which is typical for hypervirulent CC17 strains [10,26,34,35]. These two strains were isolates from pregnant women. ST-1954 was found to be a single locus variant of ST-17. ST-17 strains are hypervirulent, associated with neonatal meningitis, and account for more than 80% of GBS late-onset neonatal infections [5,6,9,10,26]. 

CC1 was the most prevalent CC in our collection of strains, consisting mainly of ST-1 isolates of serotype V. Genomic analysis demonstrated a high clonality of ST-1 and ST-14 strains belonging to CC1 (Figure 4). The overrepresentation of ST-1 with asymptomatic carriage [5,9,37], as well as the association of ST-1 with serotype V has been well-documented [8,38,39,40,41,42]. In contrast to data from Australia [43], ST-1 is a single dominant ST of serotype V in GBS strains from the US [44]. Similarly, in our study, CC1 appeared to be largely limited to ST-1, with ST-1 serotype V representing almost two-thirds of all CC1 strains. The remaining third of CC1 belonged to a cluster of ST-14 serotype VI (27%) strains in addition to one or two isolates of ST-1 with serotypes Ia, II, and VI. These rare isolates of ST1 carrying serotypes other than V may indicate potential capsular switching events [42], which has been demonstrated before for ST-1 serotype V or VI isolates [45]. 

The Alp3 protein gene was predominant among our ST-1 serotype V strains in consistence with previous publications [46,47,48,49]. However, the Epsilon protein gene dominated among ST-14 serotype VI strains. Only a single serotype VI isolate belonged to ST-1 and possessed the alpha-C protein gene. A similar observation was reported for a collection of serotype VI strains in Canada, where ST-1 isolates carried the alpha-C protein and ST-14 possessed the Epsilon protein [50]. Of note, Creti et al., 2004 [51] previously indicated the Epsilon protein as a prevalent marker in bovine strains. Unexpectedly serotype VI was a common lineage among our strains and was almost exclusively represented by ST-14 (9 out of 10 ST-14 isolates), which, according to genomic analysis, cluster closely together (Figure 4). Vaginal carriage with serotype VI is common in Japan, Malaysia, and Taiwan but infrequently reported in Europe and North America [36,52,53,54,55]. However, recent reports show the emergence of invasive serotype VI infections in Canada [50], Taiwan [56,57], and Japan [40,58]. 

Almost all of our serotype VI strains were ST-14. This was unlike the Canadian isolates where the majority of serotype VI strains belonged to ST-1 and seldom to ST-14. The Canadian strains differed from Asian serotype VI strains and had undergone recombination events in the two-component system CsrRS (or CovRS) and the Alp-encoding genes [50]. CovRS is a major regulatory system controlling a large proportion of genes that are crucial for GBS adaptation, virulence, and survival in response to environmental fluctuation [59]. For example, GBS tolerance to acid stress has been directly attributed to CovRS [60,61]. The Alp genes encode well-known GBS protein antigens. They are potential candidates for GBS vaccine formulation as well as markers for invasive infections [51,62]. Moreover, all our ST-14 isolates had a premature stop codon in the BibA immunogenic surface protein gene, a potential GBS vaccine candidate, which is tightly regulated through CovRS [61]. Through binding of BibA to the complement component 4, BibA confers an immune escape mechanism preventing complement deposition and promoting GBS survival in blood [33]. However, in other aspects, all of the ST-14 isolates presented with the molecular traits of invasive isolates, all contained the PI-1+2b pilus island profile which is typical for invasive GBS isolates [26,34], indicating that the functional BibA gene loss may have been compensated for by other virulence traits. This is in line with Santi et al., 2009 [61], who reported BibA and pilus components as pH-dependent components. They showed the upregulation of pilus components versus very low expression levels of BibA under acidic conditions. However, the molecular genetic makeup of invasive strains versus non-invasive strains is complex. Almeida and coworkers, for example, reported distinct GBS lineages in relation to CovRS mutations [63]. They observed specific and frequent mutations in CovRS being associated with invasive isolates. Genomic recombination and capsular switching drive the emergence of novel STs, which has, for example, been confirmed for serotype IV ST-459 [64] and ST-452 [65], the cluster of ST-14 strains in our collection may thus represent the emergence of a specific GBS clone. 

Interestingly, CC4 isolates represented more than 13% of our collection and were equally common to CC19. This was due to the prevalence of ST-4 and ST-932 in CC4. Of note, almost all ST-4 and ST-932 isolates (10 out 11) harbored the PI-1+2b pilus island, a characteristic feature of invasive GBS strains [34]. Sporadic ST-4 strains were previously shown in different geographic locations worldwide [5,8,11,12,22,23,46,66,67]. Several STs, however, are restricted to Africa, like, for example, ST-932 [66] from Ethiopia. The same was true for ST-934, which is also from Ethiopia [66], and was found to be positive for the HvgA adhesin. Similarly, we have one isolate of ST-934 serotype Ia in our collection, which was both positive for the HvgA protein gene and contained the PI-1+2b pilus island combination. A geographical restriction is also present in ST-569, which is the most prevalent singleton in our study. All except one ST-569 strains were isolated from pregnant women. While rarely reported elsewhere [38,43,68], ST-569 was the second most predominant ST in Ethiopia [69]. Moreover, we found one isolate of ST-486 serotype Ia, which according to the public MLST database [22], has not been detected outside of Africa. Similar geographical restrictions have been observed in Kenya for ST-484 and ST-866 [67]. Even though our strain collection is limited, it thus clearly reflects the African origin of the analyzed isolates. 

Characterizing the population structure of colonizing isolates is fundamental for understanding GBS disease and serves as a basis for vaccine development. With respect to our population, GBS trivalent and hexavalent vaccines would provide 37.7% (34/90) and 87.7% (79/90) coverage, respectively. However, region-specific lineages, as exemplified by the serotype VI isolates in our study, should be considered for inclusion to minimize the risk of the emergence of vaccine escape variants [42] or the expansion of non-vaccine serotypes [42].

## 5. Conclusions

The current study describes the MLST types of colonizing GBS in Egypt among pregnant and nonpregnant women. Although the population structure of Egyptian GBS strains resembles that of sequence types and serotypes found in the US and Europe, we identified local specificities in the distribution, including a rather substantial proportion of ST-14 serotype VI strains. These findings may have potential implications for vaccine development. Further epidemiological studies characterizing GBS colonization in Egypt and other African countries are important for a detailed description of the GBS population structure in view of future vaccine implementation.

## Figures and Tables

**Figure 1 microorganisms-11-00038-f001:**
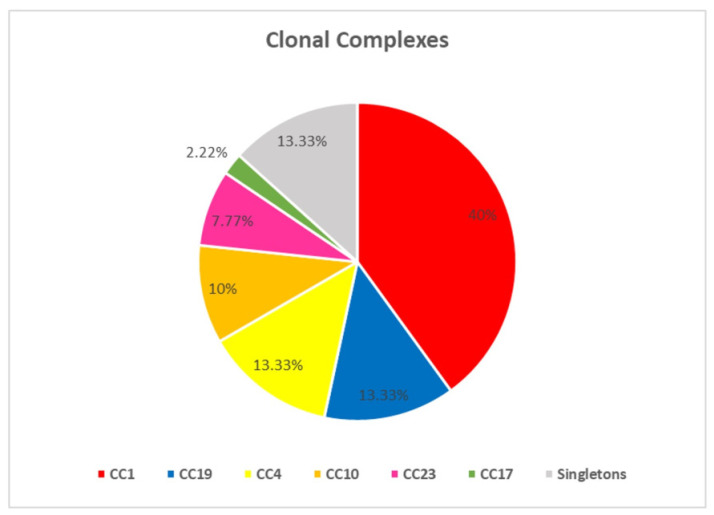
Distribution of clonal complexes among colonizing GBS strains in Egypt.

**Figure 2 microorganisms-11-00038-f002:**
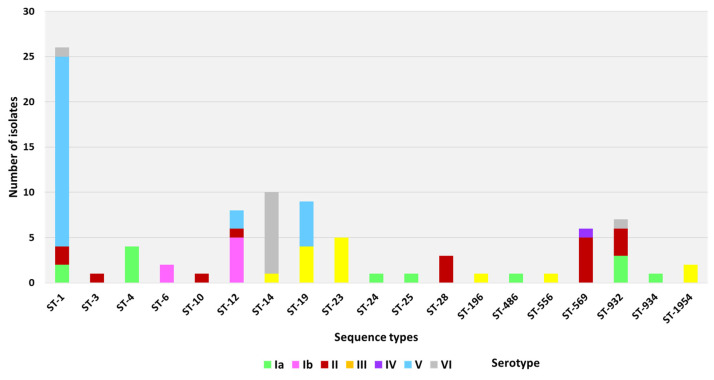
Sequence type (ST) and serotype distribution of colonizing GBS isolates from Egypt. Bars indicate the number of isolates per ST in the collection of 90 isolates. The colored blocks indicate the serotype.

**Figure 3 microorganisms-11-00038-f003:**
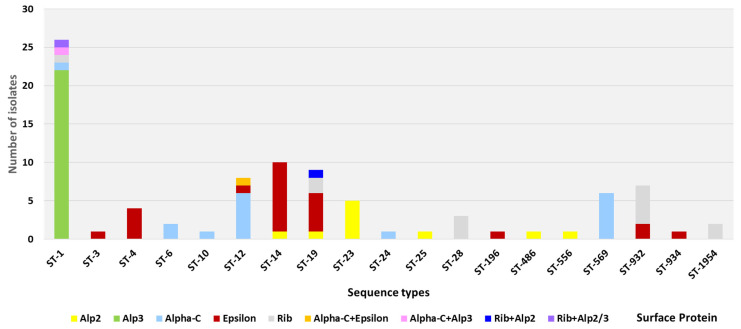
Sequence type (ST) and Alpha-like surface protein distribution of colonizing GBS isolates from Egypt. Bars indicate the numbers of isolates per ST in the collection of 90 isolates. The colored blocks indicate the surface protein.

**Figure 4 microorganisms-11-00038-f004:**
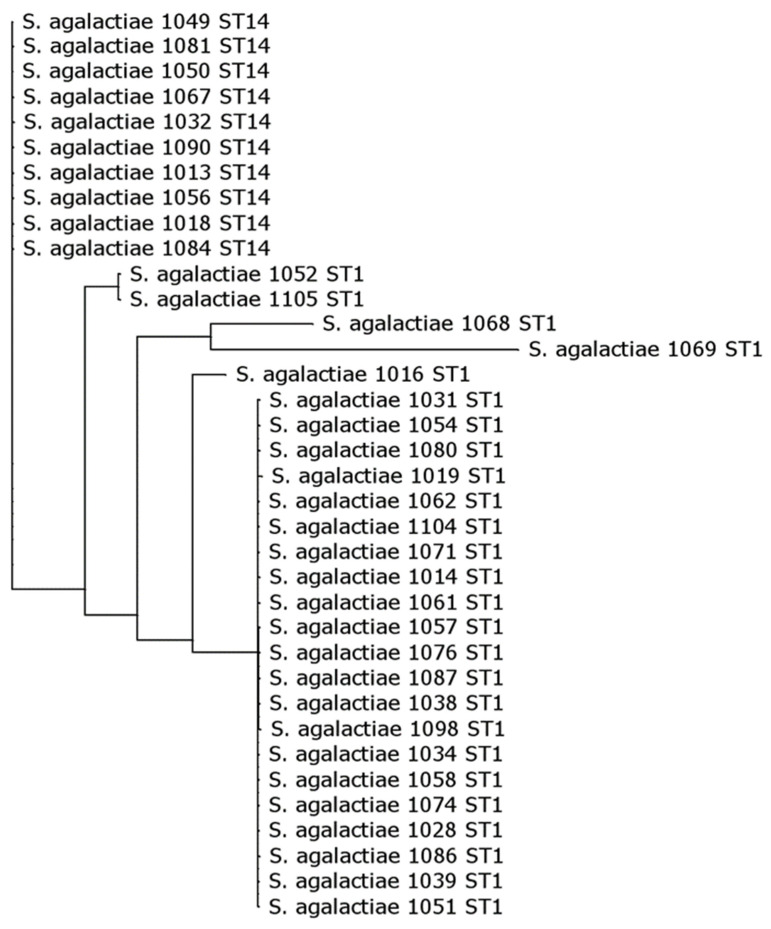
Phylogenetic analysis of GBS genomes belonging to the CC1 complex using Bowtie 2 sequence alignments.

**Table 1 microorganisms-11-00038-t001:** Serotypes, Pilus islands, Srr proteins, HvgA, and MLST distribution of colonizing GBS isolates from Egypt.

Clonal Complex/Sequence Type (Number)	Serotypes							Pilus Islands			Srr Proteins		HvgA
	Ia	Ib	II	III	IV	V	VI	1+2A	1+2B	2A	Srr1	Srr2	
**CC1**													
ST-1 (26)	2		2			21	1	24	2		25		
ST-14 (10)				1			9		10		10		
**CC4**													
ST-3 (1)			1					1			1		
ST-4 (4)	4								4		4		
ST-932 (7)	3		3				1	1	6		6		
**CC10**													
ST-10 (1)			1					1			1		
ST-12 (8)		5	1			2		8			8		
**CC17**													
ST-1954 (2)				2					2			2	2
**CC19**													
ST-19 (9)				4		5		9			9		
ST-28 (3)			3					3			1		
**CC23**													
ST-23 (5)				5				5			5		
ST-25 (1)	1							1			1		
ST-556 (1)				1				1			1		
**Singletons**													
ST-6 (2)		2								2	2		
ST-24 (1)	1									1	1		
ST-196 (1)				1				1			1		
ST-486 (1)	1								1		1		
ST-569 (6)			5		1			6			6		
ST-934 (1)	1								1		1		1
**Total**	13	7	16	14	1	28	11	61	26	3	84	2	3

**Table 2 microorganisms-11-00038-t002:** Alpha-like surface proteins and MLST distribution of colonizing GBS isolates from Egypt.

Clonal Complex/Sequence Type (Number)	Alpha-Like Surface Proteins								
	Alp2	Alp3	Alpha-C	Epsilon	Rib	Alpha-C + Epsilon	Alpha-C + Alp3	Rib + Alp2	Rib + Alp2/3
**CC1**									
ST-1 (26)		22	1		1		1		1
ST-14 (10)	1			9					
**CC4**									
ST-3 (1)				1					
ST-4 (4)				4					
ST-932 (7)				2	5				
**CC10**									
ST-10 (1)			1						
ST-12 (8)			6	1		1			
**CC17**									
ST-1954 (2)					2				
**CC19**									
ST-19 (9)	1			5	2			1	
ST-28 (3)					3				
**CC23**									
ST-23 (5)	5								
ST-25 (1)	1								
ST-556 (1)	1								
**Singletons**									
ST-6 (2)			2						
ST-24 (1)			1						
ST-196 (1)				1					
ST-486 (1)	1								
ST-569 (6)			6						
ST-934 (1)				1					
**Total**	10	22	17	24	13	1	1	1	1

## Data Availability

WGS was carried out as part of the JUNO project and sequence data are publicly available in the context of this project (https://www.sanger.ac.uk/collaboration/juno-global-genomic-survey-streptococcus-agalactiae/ (accessed on 17 December 2022)). Accession numbers are listed in Appendix A.

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
