# Peer review of "Molecular Epidemiology of Group B Streptococcus Colonization in Egyptian Women"

_microorganisms, 2022, doi:10.3390/microorganisms11010038_

Round 1

Reviewer 1 Report

Shabayek et al. investigated the molecular epidemiology of GBS strains from maternal carriage in Egypt by performing whole genome sequencing (WGS) within the JUNO project. From the WGS data, MLST and serotype of all GBS were determined and genetic information about surface proteins was extracted. The authors reported for the first time the MLST types of GBS isolated from vaginal carriage in Egypt and their possible correlations with capsular serotypes and surface proteins. They identified local specificities in the distribution including a substantial proportion of ST-14 serotype VI strains and, overall, the findings may have potential implications for vaccine developments. This is an interesting and, in general, well written research article with clear goals, study parameters and procedures.

Below there are minor comments that should be considered by the authors for revision.

Lines 46-47: The statement that MLST provides a correlation between CC and capsular types - "... and shows a correlation of clonal complexes (CCs) with capsular serotypes" - needs to be referenced. 

Figure 1: I suggest that the graphic displayed in figure 1 be changed to the graphic bar format, as it allows a better visualization of the data in terms of quantity (percentage) of each group and a better resolution of the data to make comparisons between the groups.

Author Response

Lines 46-47: The statement that MLST provides a correlation between CC and capsular types - "... and shows a correlation of clonal complexes (CCs) with capsular serotypes" - needs to be referenced. 

Response: Done as recommended.

Figure 1: I suggest that the graphic displayed in figure 1 be changed to the graphic bar format, as it allows a better visualization of the data in terms of quantity (percentage) of each group and a better resolution of the data to make comparisons between the groups.

Response: The percentages are both shown within text and in figure to make it easier for making comparisons between groups.

Reviewer 2 Report

In the manuscript titled “Molecular epidemiology of Group B Streptococcus colonization in Egypt” authors have claimed to report for the first time the MLST types of GBS  isolated from vaginal carriage in Egypt and their possible correlations with capsular serotypes and surface proteins.

Whole genome sequencing of the selected isolates was performed on the Illumina HiSeq platform. MLST analysis was conducted and different sequence types and clonal complexes were deduced. A Novel ST-1954 also claimed by the authors.

Overall the study was well performed and the manuscript is written in a very informative and sophisticated manner.

There are a few points I would like to raise-

1- The isolates used in this study were obtained from a previous work published in 2014. And the original samples were collected in the year 2010. How is this epidemiology study from more than decade-old samples significant? How this work could help in the present scenario?

2- It would be very nice if the authors add some lines about their findings in the discussion in terms of comparison between pregnant and non-pregnant women.

Author Response

1- The isolates used in this study were obtained from a previous work published in 2014. And the original samples were collected in the year 2010. How is this epidemiology study from more than decade-old samples significant? How this work could help in the present scenario?

Response: Until now no MLST data are available on human GBS isolates from Egypt. Most of the available reports are on GBS isolates of bovine or fish origin. Literally, this is the first study. So, our results represent the first MLST records on GBS isolates from Egypt, which is important information in regard to vaccine implementation. Besides, recent publications on GBS isolates from Africa pointed out that our data fit very well within the African pattern, despite being collected some time ago. Kindly check the following publication: Group B Streptococcal Colonization in African Countries: Prevalence, Capsular Serotypes, and Molecular Sequence Types DOI: 10.3390/pathogens10121606

2- It would be very nice if the authors add some lines about their findings in the discussion in terms of comparison between pregnant and non-pregnant women.

Response: Done as recommended. Kindly, check lines 289-292, 303-304, 368-369

Reviewer 3 Report

microorganisms-2101918-peer-review-v1

The authors shall re-format the manuscript according to the journal style. The work presented by Sarah Shabayek et al. titled “Molecular epidemiology of Group B Streptococcus colonization in Egypt” is of interest but should be revised by English editing professionals.

For example, the title should be: Molecular epidemiology of Group B Streptococcus colonization in Egyptian women”.

Several errors are present in the abstract, while fewer along the text:

e.i The data reports MLST data and surface protein profiles for the first time for GBS strains (isolated by what?) from Egypt. The molecular epidemiology of these strains clearly points to the African origin with the detection of several sequence types (STs) that have only been observed in Africa

After these minor revisions, the paper can be accepted. 

Author Response

The authors shall re-format the manuscript according to the journal style. The work presented by Sarah Shabayek et al. titled “Molecular epidemiology of Group B Streptococcus colonization in Egypt” is of interest but should be revised by English editing professionals.

For example, the title should be: Molecular epidemiology of Group B Streptococcus colonization in Egyptian women”.

Response: Done as Recommended

Several errors are present in the abstract, while fewer along the text:

e.i The data reports MLST data and surface protein profiles for the first time for GBS strains (isolated by what?) from Egypt. The molecular epidemiology of these strains clearly points to the African origin with the detection of several sequence types (STs) that have only been observed in Africa

After these minor revisions, the paper can be accepted. 

Response: The manuscript has been checked by the coauthors who are English natives.

Reviewer 4 Report

general comments:

the manuscript provides novel data however some language rephrasing and modification are required as suggested next. please use my next recommendations for the entire test to modify the style of your writing and presentation.

the title does generalize the article outcome. please modify it according to the aims including the source of your strains.

please delete the many use of "first" in the article. you may limit use it to the end of the abstract-conclusion.

abstract:

line18: delete "The data reports for the first time MLST data and surface protein profiles for GBS strains from Egypt"

line19: move the following into the conclusion lines in the end of the abstract and limit your results to your results: "The molecular epidemiology of these strains clearly points to the African origin with the detection of several sequence types (STs) that have only been observed in Africa."

line21: delete "Moreover". authors should limit the description in the abstract and get rid of "here, overall....etc". make the abstract as concise and possible.

Introduction

line30: please add reference to first line in intro.

line31: relocate and emerge "it may also cause serious infections in pregnant and nonpregnant adults [1,2]" into line 35-39 "GBS is a ....etc" and create a better logic lines. you may consider: GBS is a groups of opportunistic ......etc" 

emerge second paragraph (line40) and third paragraph (line48) into one concise paragraph. make it simple and reduced.

line40: move the following to appropriate location: "The population structure of GBS can vary between geographic regions [2], a major aspect to be considered in vaccine development." you may emerge into 65.

line62-65: seems not clear. The start of the the last paragraph is not effective and need to be modified. you can adopt the following:

Screening and intrapartum antibiotic prophylaxis (IAP) are recommended effective therapy for pregnant women GBS carriers however such preventive strategy are rarely adopted in the underdeveloped region particularly Africa (references). alternative therapeutic option have been introduced including .......etc"

line72 -77: need to be rewritten and modified. this summarize the aims of the study which is quite clear and can be singular general aim. e.g. molecular epidemiology of ...... isolates from .... between ?????. also, the correlation of ??? with ??? was also investigated.

material and methods:

this section could be divided into 2 subheadings to follow the results flow. i am not sure if MLST description should be described first to follow also the results sequence.

line79: this could be slightly modified and moved to aims and specify the total number of isolates "GBS isolates were collected from pregnant and nonpregnant women in an earlier investigation [25]."

line88: "BURST analysis ...etc" might be better moving the reference to the end of the sentence 

Results

line102-105: organise these two sentences into one good summary results. for example:

"A total of 19 unique MLST STs were identified of which ????? followed respectively by .......etc"

line110-113: should be limited to what you have found and only mention and discuss such revelation in the discussion.

line143-145: these should be in introduction or discussion. 

line202-203: rephrase please. do not narrate and make it concise. for instance, you dont need to say "defined here"

line 204-206: emerge these two sentence. for example:

Among the ST-1 and ST-14 strains, ST-1/V/Alp3 (n=20/26) and ST-205 14/VI/Epsilon (n=9/10) were respectively the predominant lineages.

line 206: rephrase please as follows: 

"further genomic analysis showed .......etc"

please use the above suggested (indirect) style to rephrase most of the following lines in results.

Discussion

line 231: please deleted "Here we report for the first time". 

in the first paragraph you mention the important of your results in vaccine development. please discuss your results first then discuss the importance in vaccine development.

line 235: discuss your results then say the consistency with other reports. please rephrase into better concise language. similarly for line 258 and line 271.

i don't digest "Dissimilar to our results" in line 271 so delete and rephrase as suggested above.

line 297: rephrase please and i suggest "interestingly".

line 319: rephrase please and use passive instead of " we ...etc" you provide novel data and information so focus on that as well and limit the use of "first time" please. you can use is in conclusion-abstract only.

Author Response

Detailed Response to Reviwer-4

Comments and Suggestions for Authors

general comments:

the manuscript provides novel data however some language rephrasing and modification are required as suggested next. please use my next recommendations for the entire test to modify the style of your writing and presentation.

the title does generalize the article outcome. please modify it according to the aims including the source of your strains.

Response: Done as recommended. We have changed the title as recommended by Reviewer 3.

please delete the many use of "first" in the article. you may limit use it to the end of the abstract-conclusion.

Response: Done as recommended.

abstract:

line18: delete "The data reports for the first time MLST data and surface protein profiles for GBS strains from Egypt"

Response: Done as recommended.

line19: move the following into the conclusion lines in the end of the abstract and limit your results to your results: "The molecular epidemiology of these strains clearly points to the African origin with the detection of several sequence types (STs) that have only been observed in Africa."

Response: Done as recommended.

line21: delete "Moreover". authors should limit the description in the abstract and get rid of "here, overall....etc". make the abstract as concise and possible.

Response: Done as recommended.

Introduction

line30: please add reference to first line in intro.

Response: Done as recommended.

line31: relocate and emerge "it may also cause serious infections in pregnant and nonpregnant adults [1,2]" into line 35-39 "GBS is a ....etc" and create a better logic lines. you may consider: GBS is a groups of opportunistic ......etc" 

Response: Done as recommended.

emerge second paragraph (line40) and third paragraph (line48) into one concise paragraph. make it simple and reduced.

Response: Done as recommended.

line40: move the following to appropriate location: "The population structure of GBS can vary between geographic regions [2], a major aspect to be considered in vaccine development." you may emerge into 65.

Response: Done as recommended.

line62-65: seems not clear. The start of the the last paragraph is not effective and need to be modified. you can adopt the following:

Screening and intrapartum antibiotic prophylaxis (IAP) are recommended effective therapy for pregnant women GBS carriers however such preventive strategy are rarely adopted in the underdeveloped region particularly Africa (references). alternative therapeutic option have been introduced including .......etc"

 Response: Done as recommended.

line72 -77: need to be rewritten and modified. this summarize the aims of the study which is quite clear and can be singular general aim. e.g. molecular epidemiology of ...... isolates from .... between ?????. also, the correlation of ??? with ??? was also investigated.

 Response: Done as recommended.

material and methods:

this section could be divided into 2 subheadings to follow the results flow. i am not sure if MLST description should be described first to follow also the results sequence.

Response: Done as recommended.

line79: this could be slightly modified and moved to aims and specify the total number of isolates "GBS isolates were collected from pregnant and nonpregnant women in an earlier investigation [25]."

Response: Done as recommended.

line88: "BURST analysis ...etc" might be better moving the reference to the end of the sentence 

Response: Done as recommended.

Results

line102-105: organise these two sentences into one good summary results. for example:

"A total of 19 unique MLST STs were identified of which ????? followed respectively by .......etc"

Response: Done as recommended.

line110-113: should be limited to what you have found and only mention and discuss such revelation in the discussion.

Response: Done as recommended.

line143-145: these should be in introduction or discussion. 

Response: Done as recommended.

line202-203: rephrase please. do not narrate and make it concise. for instance, you dont need to say "defined here"

Response: Done as recommended.

line 204-206: emerge these two sentence. for example:

Among the ST-1 and ST-14 strains, ST-1/V/Alp3 (n=20/26) and ST-205 14/VI/Epsilon (n=9/10) were respectively the predominant lineages.

Response: Done as recommended.

line 206: rephrase please as follows: 

"further genomic analysis showed .......etc"

Response: Done as recommended.

please use the above suggested (indirect) style to rephrase most of the following lines in results.

Response: Done as recommended.

Discussion

line 231: please deleted "Here we report for the first time". 

Response: Done as recommended.

in the first paragraph you mention the important of your results in vaccine development. please discuss your results first then discuss the importance in vaccine development.

Response: Done as recommended.

line 235: discuss your results then say the consistency with other reports. please rephrase into better concise language. similarly for line 258 and line 271.

Response: Done as recommended.

i don't digest "Dissimilar to our results" in line 271 so delete and rephrase as suggested above.

Response: Done as recommended.

line 297: rephrase please and i suggest "interestingly".

Response: Done as recommended.

line 319: rephrase please and use passive instead of " we ...etc" you provide novel data and information so focus on that as well and limit the use of "first time" please. you can use is in conclusion-abstract only.

Response: Done as recommended.
